# CFT-RAG: An Entity Tree Based Retrieval Augmented Generation Algorithm With Cuckoo Filter

**Zihang Li, Wenjun Liu, Zhengyang Wang & Tong Yang** *
State Key Laboratory of Multimedia Information Processing
School of Computer Science
Peking University
Beijing, China
{lizihang@stu.pku.edu.cn, yangtong@pku.edu.cn}

**Yangdong Ruan**
School of Software
Beijing University of Aeronautics and Astronautics
Beijing, China

## Abstract

Although retrieval-augmented generation(RAG) significantly improves generation quality by retrieving external knowledge bases and integrating generated content, it faces computational efficiency bottlenecks, particularly in knowledge retrieval tasks involving hierarchical structures for Tree-RAG. This paper proposes a Tree-RAG acceleration method based on the improved Cuckoo Filter, which optimizes entity localization during the retrieval process to achieve significant performance improvements. Tree-RAG effectively organizes entities through the introduction of a hierarchical tree structure, while the Cuckoo Filter serves as an efficient data structure that supports rapid membership queries and dynamic updates. The experiment results demonstrate that our method is much faster than baseline methods while maintaining high levels of generative quality. For instance, our method is more than 800% faster than naive Tree-RAG on DART dataset. Our work is available at https://github.com/TUPYP7180/CFT-RAG-2025.

## 1 Introduction

In the era of information explosion, Retrieval-Augmented Generation (RAG), a technology integrating retrieval mechanisms with generative models, has gained significant attention. It allows models to draw on external knowledge bases during text generation, effectively overcoming the limitations of traditional generative models in knowledge-intensive tasks (Lewis et al., 2020). The knowledge base, a vital part of the RAG system, stores a wealth of structured and unstructured knowledge, acting as the main source of external information for the model. However, with the continuous expansion of the knowledge base and the rapid pace of knowledge update, the challenge of efficiently retrieving relevant and accurate information from it has become a major obstacle to improving the performance of RAG system. The experiment results of time ratio in Table 1 show that the retrieval time accounts for 10% to 72% of the total response time in all baseline RAG methods. Enhancing the retrieval speed and accuracy of the knowledge base is crucial for boosting the overall performance of the RAG system (Zhong et al., 2024). Faster and more accurate retrieval enables the model to access relevant knowledge promptly, improving response speed and the quality of generated content. In contrast, inefficient or inaccurate retrieval can result in incorrect or irrelevant outputs, degrading user experience and system usability. Therefore, exploring ways to optimize the knowledge base retrieval

---

*Corresponding Author.

mechanism is of great theoretical and practical importance, and this paper will focus on this key issue.

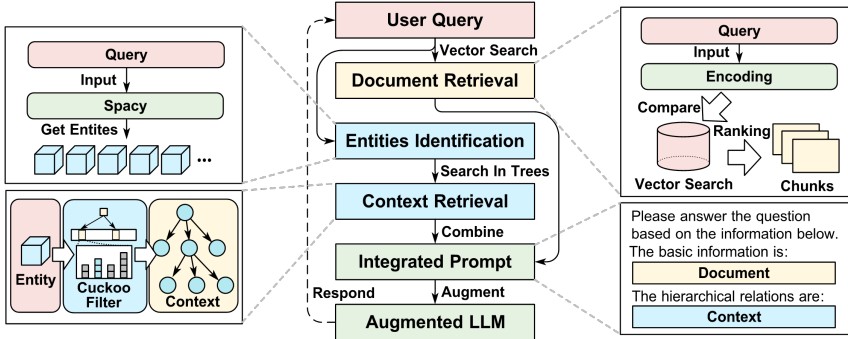

Figure 1: The workflow of CFT-RAG begins with a user query, which undergoes vector search to retrieve relevant documents. Key entities are then identified from entity trees by applying SpaCy and hierarchical tree searches. Context information related to these entities is retrieved and filtered efficiently by applying Cuckoo Filter. The retrieved context and hierarchical relationships are integrated into a comprehensive prompt, which is fed into an augmented large language model (LLM). The LLM processes this enriched prompt to generate a context-aware and accurate response to the user query.

Knowledge bases in Retrieval-Augmented Generation (RAG) systems are mainly of three types: text-based, graph-based, and tree-based (Han et al., 2025). Text-based ones store information as text, easy to manage but slow in retrieval due to complex language processing. Graph-based knowledge bases represent knowledge as graphs, excelling in handling complex relationships with relatively fast retrieval for certain queries, thanks to graph neural networks. Tree-based knowledge bases structure knowledge hierarchically. Despite text-based and graph-based knowledge bases having made good progress, the retrieval speed of all three types, especially tree-based ones, needs improvement. For RAG systems to provide faster and more accurate responses, optimizing retrieval from these knowledge bases, particularly tree-RAG, is a key research challenge.

Tree-RAG, an extension of RAG, improves on traditional RAG frameworks by using a hierarchical tree structure to organize the retrieved knowledge, thus providing richer context and capturing complex relationships among entities. In Tree-RAG, entities are arranged hierarchically, allowing the retrieval process to more effectively traverse related entities at multiple levels. This results in enhanced response accuracy and coherence, as the tree structure maintains connections between entities that are essential for contextually rich answers (Fatehkia et al., 2024). However, a critical limitation of Tree-RAG lies in its computational inefficiency: as the datasets and tree depth grow, the time required to locate and retrieve relevant entities within the hierarchical structure significantly increases, posing scalability challenges. This paper aims to greatly improve the retrieval efficiency of Tree-RAG without sacrificing the accuracy of the generated responses.

To address the retrieval bottleneck, we introduce an optimized method to Tree-RAG by integrating the Cuckoo Filter (Fan et al., 2014), a high-performance data structure. Its basic workflow is presented in Figure 1. The Cuckoo Filter excels in fast membership queries and supports dynamic operations, such as element insertions and deletions, making it suitable for dynamic data management scenarios (Fan et al., 2014). Unlike traditional filters, such as Bloom Filter (Guo et al., 2010), which is limited by fixed false-positive rates and lack deletion capability, the Cuckoo Filter allows for both flexible updates and reduced storage requirements. Therefore, it is particularly advantageous for handling large hierarchical datasets. The comparison experiment is designed to demonstrate that our method is significantly better than the three baseline methods including naive Tree-RAG.

Theoretically, the time complexity of Cuckoo Filter for searching entities is O(1), which is significantly lower than that of naive Tree-RAG. From a spatial point of view, entities are stored in the Cuckoo Filter in the form of fingerprints (12-bit), which greatly saves memory usage. On the other hand, when the load factor of Cuckoo Filter exceeds the preset threshold, the storage capacity is usually

increased by double expansion, while the original elements are re-hashed and migrated to the new storage location to complete the automatic expansion. This keeps the loading rate of cuckoo filter high and not too high, thus saving memory while avoiding hash collisions as much as possible.

Moreover, we propose two novel designs. The first design introduces a temperature variable, with each entity stored in the Cuckoo Filter maintaining an additional variable called temperature. The variable is used to record the frequency of the entity being accessed. The Cuckoo Filter sorts the entities according to the frequency, and the entities with the highest temperature are placed in the front of the bucket, thus speeding up the retrieval. The second design is to introduce block linked list, where Cuckoo Filter stores the addresses of entities at different locations in the tree. The utilization of the space of block linked list is high, it can support relatively efficient random access, reduce the number of linked list nodes, and perform well in balancing time and space complexity, especially for processing large-scale data. Therefore, we achieve acceleration by storing these addresses in the form of a block linked list. An ablation experiment is performed to demonstrate the effectiveness of the design.

## 1.1 RELATED WORK

**Cuckoo Filter** Cuckoo Filter is an efficient data structure for supporting fast element lookup and deletion operations, which is mainly used for collection operations and data stream processing (Fan et al., 2014). It is developed based on the idea of Cuckoo Hashing (Pagh & Rodler, 2001). Cuckoo Filter outperforms traditional Bloom filters in terms of storage efficiency and query performance, especially in scenarios where frequent insertion and deletion of elements are required. The main advantage of Cuckoo Filter is its support for dynamic updates, which enables it to efficiently handle element changes in a collection. Unlike Bloom Filter, Cuckoo Filter can not only query whether an element exists or not, but also support the deletion operation of an element, a feature that is important in many practical applications (Gupta & Breitinger, 2015). The working principle of Cuckoo Filter is based on multiple hash functions and a bucket structure, which is utilized by storing the elements in fixed-size buckets and using the hash conflicts to achieve fast lookup of elements. This efficient data structure provides new ideas for handling large-scale datasets, which can accelerate the retrieval process and improve response efficiency.

**Retrieval Augmented Generation** Retrieval Augmented Generation(RAG) is a state-of-the-art method that combines information retrieval with large language models, with the aim of addressing the limitations of it in the absence of specific knowledge (Lewis et al., 2020). The core idea of RAG is to utilize an external knowledge base for retrieval and to incorporate relevant information into the generation process. Specifically, RAG first retrieves multiple relevant documents from the knowledge base on the basis of the input query, and then combines the retrieved knowledge with the input query to form the context and then generate the final answer through the generative model. This approach significantly improves the quality and relevance of the generated text (Karpukhin et al., 2020) and avoids the limitations of fixed model knowledge (Gupta et al., 2024). Compared to fine-tuning, RAG has a greater ability to update knowledge and also reduces the dependence on large-scale data. Moreover, CRUD-RAG (Lyu et al., 2024) provides a Chinese benchmark covering four main application scenarios of RAG. MemoryBank (Zhong et al., 2024) enhances long-term memory for LLMs, improving recall and user adaptation. RAPTOR (Sarthi et al., 2024) employs a tree-structured retrieval approach with recursive embedding, clustering, and summarization, excelling in semantic understanding. These works emphasize relationship extraction and dynamic updating but lack strong retrieval efficiency across diverse applications. CFT-RAG addresses these gaps with Cuckoo Filter.

**Graph-RAG** Graph Retrieval-Augmented Generation (Graph-RAG) is an extension of RAG, where the information retrieval process is augmented by leveraging graph-based structures to organize and retrieve information (Lewis et al., 2020). The key difference between traditional RAG and Graph-RAG is the use of a graph, such as a knowledge graph, to model relationships between entities and concepts, which can improve the relevance and contextuality of the retrieved information (Hu et al., 2024). The algorithmic process of Graph-RAG enhances the understanding of relationships and provides more precise information retrieval than traditional RAG (Darrin et al., 2024). However, the complexity in graph construction and maintenance will be a trouble, the quality and completeness of the graph can also affect the accuracy of responses generated by the model. For instance, EMG-RAG (Wang et al., 2024) integrates Editable Memory Graph for partition management and relation capture,

improving answer quality but suffering from high computational cost. Therefore, in comparison to Tree-RAG (Fatehkia et al., 2024), Graph-RAG still has certain disadvantages.

**Tree-RAG** Tree-RAG(T-RAG) is an emerging method that combines tree structure and large language models to improve the effectiveness of knowledge retrieval and generation tasks. Compared to traditional RAG, T-RAG further enhances the context retrieved from vector databases by introducing a tree data structure to represent the hierarchy of entities in an organization. The algorithmic process of Tree-RAG consists of the following steps: first, the input query is parsed to identify relevant entities and the retrieval of relevant entities is performed in the constructed forest. Next, the system traverses through the hierarchical structure of the tree to obtain the nodes related to the query entity and its upper and lower multilevel parent-child nodes. Subsequently, the retrieved knowledge is fused with the query to generate the augmented context. Finally, the generative model generates the final answer based on the augmented context. This process effectively combines knowledge retrieval and generation and improves the accuracy and contextual relevance of the generative model (Fatehkia et al., 2024). However, T-RAG runs inefficiently due to the time-consuming nature of finding all the locations of related entities in a forest with a large amount of data. Our method applies the improved Cuckoo Filter to the retrieval process of Tree-RAG, making it greatly faster.

## 2 DATA PRE-PROCESSING

It is important to recognize entities and construct hierarchical relationships (e.g., tree diagrams) between entities from datasets. It mainly involves the steps of entity recognition, relationship extraction and filtering. For existing hierarchical data, binary pairs representing parent-child relationships are directly extracted. For raw textual data, text cleansing is first performed manually to remove irrelevant information.

### 2.1 ENTITIES RECOGNITION

SpaCy is a Python library, and its entity recognition function is based on deep learning models (e.g., CNN and Transformer). It captures information by transforming the text into word vectors and feature vectors. The models are trained on a labeled corpus to recognize named entities in the text, such as names of people and places. We adopt the method in T-RAG by using the spaCy library to recognize and extract entities from a user's query (Fatehkia et al., 2024).

### 2.2 RELATIONSHIP EXTRACTION

Various relationships are identified from the data, including organizational, categorization, temporal, geographic, inclusion, functional, and attribute relationships. The relationships manifest through grammatical structures such as noun phrases, prepositional phrases, relative clauses, and appositive structures (Vaswani et al., 2017; Devlin et al., 2019). We focus on extracting relationships that express dependency, such as "belongs to," "contains," and "is dependent on." The process is presented in Figure 2.

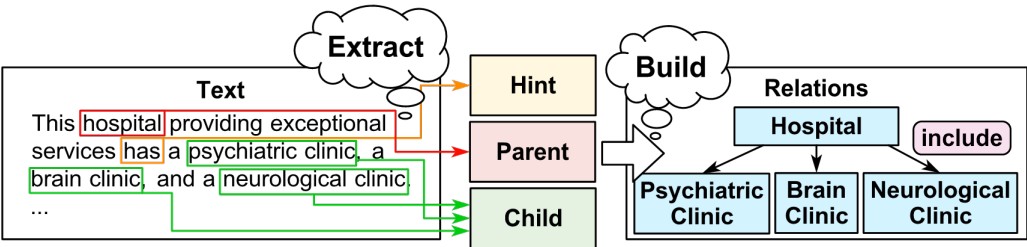

Figure 2: The process of relationship extraction

We apply several dependency parsing models(gpt-4 and open-source NLP libraries) to analyze the grammatical structure of the data. This helps identify relationships between words, such as subject-verb-object or modifier relationships.

We define rules to identify hierarchical relationships. If a word modifies another noun, it can be interpreted as a child-parent relationship; If there are conjunctions (e.g., "and", "or"), handle them to group entities under the same parent. As a result, there is a list of tuples representing the hierarchical structure.

## 2.3 RELATIONSHIP FILTERING

After extracting relations, certain relationships are filtered out to ensure maintain the tree structure:

- Transitive Relations: If transitive relations are detected (e.g., "A belongs to B", "B belongs to C" and "A belongs to C"), remove distant relations.
- Cycle Relations: If cycles are detected (e.g., "A belongs to B" and "B belongs to A"), only the closest relationship is retained.
- Self-Pointing Edges: Any relation where a node points to itself is removed.
- Duplicate Edges: Multiple edges between the same nodes are pruned, leaving only one edge.

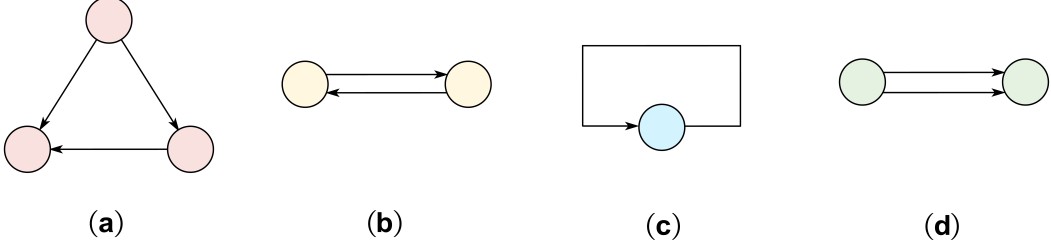

(**a**)          (**b**)          (**c**)          (**d**)

Figure 3: Error relation examples.

## 3 METHODOLOGY

In this section, we propose a novel design of Cuckoo Filter that combines the advantages of traditional Cuckoo Hashing and applies it to Tree-RAG by introducing additional designs that greatly improve the speed of knowledge retrieval in Tree-RAG.

## 3.1 STORAGE MODE

In addition to entity trees, we set up an additional Cuckoo Filter to store some entities to improve retrieval efficiency. Based on the naive Cuckoo Filter, we introduce the block linked list for optimization, which can greatly reduce memory fragmentation. We first find out all locations of each entity in the forest and then store these addresses in a block linked list.

To further optimize the retrieval performance, we propose an adaptive sorting strategy to reorder the entities in each bucket in the Cuckoo Filter based on the temperature variable which is stored at the head of the block list. The temperature variable records how often each entity is accessed, and entities with high-frequency access are prioritized to be placed at the front of the bucket. Since the Cuckoo Filter looks up the elements in the buckets linearly, this reordering mechanism can significantly optimize the query process, which can further improve the response speed of the model. In summary, in each entry of the bucket, an entity's fingerprint, its temperature, and head pointer of its block linked list are stored. The storage mode is included in Figure 4 and details about the process of insertion and eviction of entities are provided in Appendix A.

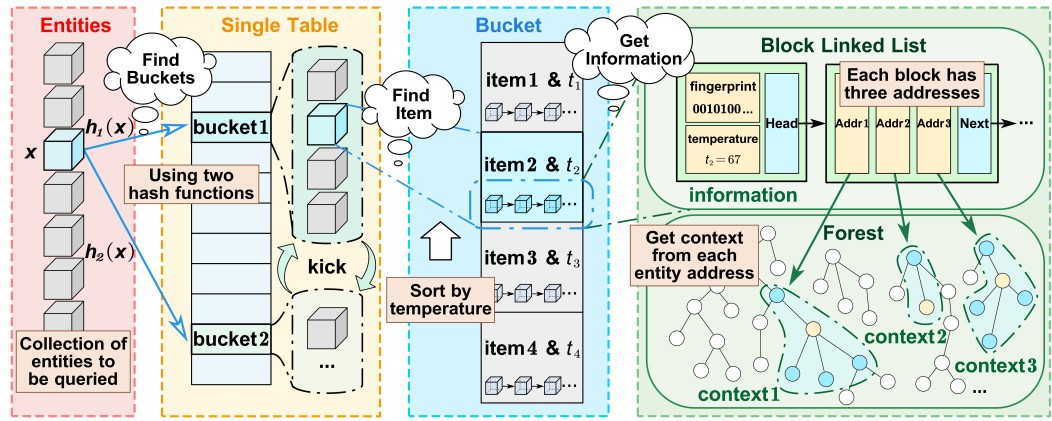

Figure 4: The workflow of CFT-RAG when query contains entity x. The entity with high temperature will be placed ahead of which with low temperature in the bucket. All the addresses in different trees of the entity are linked by the block linked list.

## 3.2 CONTEXT GENERATION

After the fingerprint of the target entity is found, the temperature of the entity is added by one and a pointer to the head of the corresponding block linked list of that entity is returned. From this pointer, the location of the entity node in different trees including multi-level parent nodes, child nodes, etc. can be accessed through the address stored in the block list. If no matching fingerprint is found, the null pointer is returned. For the queried entity and its parent and child nodes in different trees, we form a context between the entity and its relevant nodes based on the set template. For instance, the upward hierarchical relationship of entity A are: B, C and D. Finally, we fuse this information with the query to generate the augmented context. After that, the augmented context combined with system prompt and query is regarded as the prompt. The lookup and context generation process is stated in Figure 4.

---

**Algorithm 1:** Context Generation Algorithm

---

**Input:** $x$: Input entity
**Output:** $context$: Context generated for entity $x$
$f(x) \leftarrow \text{fingerprint}(x)$;
**if** *bucket[$i_1$] or bucket[$i_2$] contains $f(x)$* **then**
    $temperature \leftarrow temperature + 1$;
    **return** *head*;
**end**
$currentBlock \leftarrow \text{head} \rightarrow \text{next}$;
**while** $currentBlock \neq NULL$ **do**
    **foreach** *location in currentBlock* **do**
        Let $loc$ be the current location of entity $x$ in the block;
        Find the set of hierarchical relationship nodes at location $loc$ in the tree;
        $H_{up} \leftarrow \{h_1, h_2, \ldots, h_n\}$;
        $H_{down} \leftarrow \{h'_1, h'_2, \ldots, h'_n\}$;
        Record the first $n$ upward and downward hierarchical relationship nodes;
        **for** *($i = 1$ ; $i < n$ ; $i$++)* **do**
            Store $(h_i, h'_i)$ in context;
        **end**
    **end**
    $currentBlock \leftarrow currentBlock \rightarrow \text{next}$;
**end**
**for** *($i = 1$ ; $i < n$ ; $i$++)* **do**
    $context \leftarrow context \cup (h_i, h'_i)$;
**end**

---

# 4 EXPERIMENTS

In this section, we describe our experimental setup, baseline methods, and CFT-RAG, and comprehensively evaluate the results. Our experiments aim to assess the efficiency of CFT-RAG approach under diverse experimental conditions, particularly in terms of retrieval speed and computational overhead.

## 4.1 BASELINE

To benchmark the effectiveness of the proposed Cuckoo Filter T-RAG, we compare it against several baseline models. Our baselines include the text-based RAG, Naive T-RAG without filtering mechanisms, T-RAG with ANN and Graph-RAG with ANN. These baselines allow us to quantify the improvements introduced by Cuckoo Filter T-RAG.

**Naive T-RAG** This basic implementation of T-RAG (Fatehkia et al., 2024) does not include any filtering optimizations. The method constructs an entity tree using entities extracted from the dataset and employs a Breadth-First Search (BFS) algorithm for entity lookup. Although this approach has high time complexity and prolonged search time, it provides a straightforward baseline for evaluating the benefits of incorporating filtering mechanisms.

**Text-based RAG** This baseline RAG model retrieves the top-K relevant documents from a vector store based purely on similarity scores, without considering entity relationships or structural information. It performs standard dense retrieval followed by response generation using a sequence-to-sequence model. Although simple and easy to implement, this approach may suffer from irrelevant or redundant information due to lack of context-aware filtering or reasoning capabilities. It serves as a foundational benchmark to assess the advantages of incorporating more structured or semantic retrieval strategies.

**ANN Graph RAG** This model integrates approximate nearest neighbor (ANN) search with a graph-based entity structure to accelerate retrieval while maintaining semantic relevance. Entities and their relationships are organized into a directed graph, enabling multi-hop traversal and contextual inference. During retrieval, ANN is used to identify the closest matching entities efficiently, and the graph structure guides the expansion to related entities for enriched context. This method balances speed and accuracy by leveraging the fast lookup capabilities of ANN and the expressive power of graph reasoning.

**ANN Tree-RAG** In this variant, Approximate Nearest Neighbor (ANN) search is employed to accelerate document retrieval in the entity tree structure. Instead of performing exact similarity search, the model leverages efficient ANN indexing techniques (e.g., FAISS or HNSW) to retrieve top-K candidates for each entity. ANN T-RAG provides a strong balance between performance and efficiency, especially compared to the Naive T-RAG baseline.

## 4.2 CFT-RAG

Cuckoo Filter supports entity deletion operation, which is suitable for ongoing data update, and it has a lower false positive rate and is more space efficient. The CFT-RAG method stores the individual nodes of the entity in the forest in each bucket of the Cuckoo Filter, i.e. it merges the Cuckoo Hash with the Cuckoo Filter. After the entity tree is generated, the nodes with the same entity details in each tree are concatenated into a block list, where the pointer to the head of the list corresponds to the fingerprint, and stored together in buckets.

An improved CFT-RAG is to maintain access popularity of each entity, called temperature, at the head node of each block list, and raise the level of temperature corresponding to the hit entity during retrieval. For each bucket, if there is a bucket that has not been searched, i.e. if it is free, the fingerprints and block list header pointers in the bucket can be sorted according to temperature, and the fingerprints with higher access popularity are placed at the front of the bucket, which can take advantage of the locality of the entities contained in the user questions to improve the running speed of the algorithm.

## 4.3 DATASETS AND ENTITY FOREST

Our experiments use three datasets: the large-scale dataset MedQA (Jin et al., 2021) and two medium-sized datasets AESLC (Zhang & Tetreault, 2019) and DART (Nan et al., 2021). We leverage dependency parsing models to extract entities and relationships among them and construct the entity forest based on these extracted entities and relationships. The resulting entity forest is structured to allow efficient retrieval and provides a practical evaluation scenario for our approach. The cost is stated in Appendix A.

## 4.4 SETUP

We implement the core RAG architecture in Python, while key data structures, including Cuckoo Filters, are optimized in C++ for performance. All experiments were conducted on a system equipped with an Nvidia H100 GPU, 22 CPU cores, and 220 GiB of memory. Each algorithm was repeated 108 times to account for variability and ensure reliable results, with averages calculated across runs to mitigate the influence of outliers. We apply the LangSmith framework to evaluate the accuracy of answers, where the OpenAI scoring model used by LangSmith was replaced with doubao (Doubao AI, 2024).

Table 1: Retrieval time and accuracy on MedQA, AESLC and DART datasets

| Dataset | Algorithm | Time(s) | Time Ratio(%) | Acc(%) |
|---|---|---|---|---|
| MedQA | Text-based RAG | - | - | $51 \pm 5$ |
| | Naive T-RAG | 19.45 | 58 | $65 \pm 5$ |
| | ANN T-RAG | 7.65 | 25 | $67 \pm 4$ |
| | ANN G-RAG | 8.78 | 26 | $61 \pm 6$ |
| | CFT-RAG | **5.24** | **16** | $69 \pm 4$ |
| AESLC | Text-based RAG | - | - | $42 \pm 6$ |
| | Naive T-RAG | 12.87 | 62 | $55 \pm 5$ |
| | ANN T-RAG | 2.52 | 13 | $56 \pm 6$ |
| | ANN G-RAG | 2.38 | 11 | $53 \pm 5$ |
| | CFT-RAG | **0.97** | **5** | $57 \pm 5$ |
| DART | Text-based RAG | - | - | $53 \pm 6$ |
| | Naive T-RAG | 16.03 | 74 | $65 \pm 5$ |
| | ANN T-RAG | 3.28 | 15 | $66 \pm 5$ |
| | ANN G-RAG | 3.95 | 19 | $65 \pm 6$ |
| | CFT-RAG | **1.81** | **9** | $68 \pm 5$ |

CFT-RAG represents Cuckoo Filter T-RAG.
ANN G-RAG represents Graph-RAG with ANN.
ANN T-RAG represents Tree-RAG with ANN.
Time represents the retrieval time.
Time Ratio represents the proportion of retrieval time in response time.
Acc represents the model's accuracy of answer.

## 4.5 RESULTS AND EVALUATIONS

### 4.5.1 COMPARISON EXPERIMENT

We conduct the experiments by selecting 1,000 questions on each dataset. Then we record the average retrieval time and average response accuracy by LangSmith. Table 1 presents the retrieval time and accuracy of various RAG-based models across the MedQA, AESLC, and DART datasets. As expected, the Naive T-RAG model incurs the highest retrieval latency due to its exhaustive BFS-based search, while offering moderate accuracy improvements over the text-based baseline. Further efficiency gains are observed in ANN-based methods (ANN T-RAG and ANN G-RAG), which leverage approximate nearest neighbor search to achieve faster response times with comparable accuracy. Notably, CFT-RAG consistently outperforms other variants by achieving the lowest retrieval time across all datasets while maintaining high accuracy, demonstrating the effectiveness of integrating probabilistic filtering with structural optimization. Moreover, when the problem is complex involving multi-hop and the

required entity relationships are precise, our method shows an obvious advantage over the other methods. Use cases are provided in Appendix A.

Moreover, the error rate of our method in the process of searching entities is very low. After building the trees based on the dataset, the Cuckoo Filter includes 1024 buckets, each of which can hold up to 4 fingerprints and block linked list head pointers. The Cuckoo Filter's own memory expansion strategy is to increase the number of buckets by a power of two. In the experimental datasets, there are thousands of entities that can be extracted, and the space load factor for the Cuckoo Filter is more than 70%. Because the space load factor is not too high and searching errors are mainly caused by hash collisions, the error rate is almost 0, showing that the number of entities causing the lookup error is 0 to 1 out of 1024 buckets for thousands of entities.

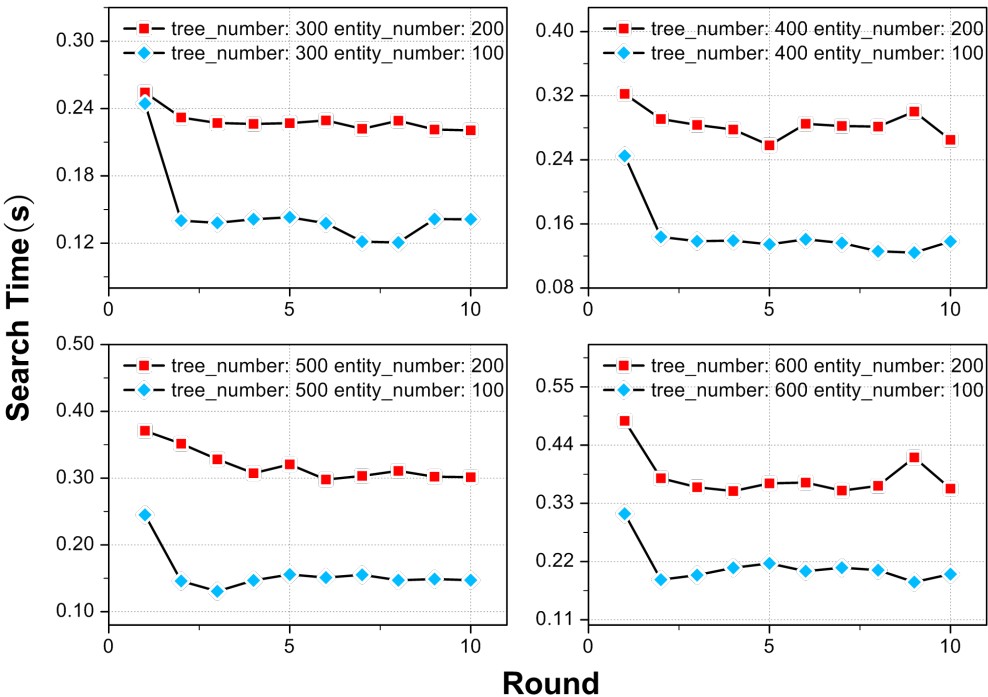

Figure 5: We record the search time per round of query with different number of trees and entities. Each round represents a search in the entities forest, and entities are inserted into the improved Cuckoo Filter before the first search is performed.

### 4.5.2 ABLATION EXPERIMENT

Sorting the fingerprints and head pointers of the block linked lists by temperature can optimize the retrieval time without occupying any extra space, which is eminently useful when the query given by the user contains a large number of entities. We design experiments to measure the effect of having or not having the sorting design on the result. In figure 5, we can observe that the retrieval time after the first round is significantly shorter than that of the first round. This is because the temperatures are updated according to the access frequency in each round and after each query, the Cuckoo Filter sorts the entities according to the entities' temperatures. This sorting design allows the 'hot' entities to be found more quickly in subsequent queries.

## 5 CONCLUSION

In this paper, we have introduced an efficient acceleration method for the Tree-RAG framework by integrating the improved Cuckoo Filter into the knowledge retrieval process. Tree-RAG, which combines hierarchical tree structures for knowledge representation with generative models, holds great promise for improving the quality and contextual relevance of generated responses. However, its performance is hindered by the computational inefficiencies of retrieving and organizing large-scale knowledge within complex tree structures.

By leveraging the Cuckoo Filter, which supports fast membership queries and dynamic updates, we have significantly enhanced the speed and efficiency of the retrieval process in Tree-RAG. Our experimental results show that the Cuckoo Filter improves retrieval times without sacrificing the quality of generated responses, making the system more scalable for real-world applications. This acceleration is particularly valuable in scenarios where real-time knowledge updates and rapid information retrieval are critical, such as in large-scale question answering, decision support systems, and conversational agents.

Future work could explore further optimizations, such as adapting the method for different knowledge structures or extending it to more complex multimodal tasks. Overall, this research demonstrates the potential of efficient data structures to enhance the performance of large language models in retrieval-intensive applications.

### ACKNOWLEDGMENTS

This work was supported by the National Key Research and Development Program of China under Grant No.2024YFB2906601, and in part by the National Natural Science Foundation of China (NSFC) (No.62372009).

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

## A    APPENDIX

### A. ENTITY INSERTION STRATEGY

The fingerprint is a shorter hash representation of an entity x, which is usually represented in fixed-length bits. Thus, we introduce fingerprints to save memory. After the tree structure is constructed, the fingerprint is first computed for each entity before inserting. Furthermore, the insertion and eviction strategy is consistent with the traditional Cuckoo Filter, where the locations of the fingerprint are determined by two hash functions Fan et al. (2014). When inserting an entity by applying the Cuckoo Filter, it first tries to store its fingerprint in the empty position $i_1$ or $i_2$, which can be calculated as:

$$i_1 = h(x), i_2 = i_1 \oplus h(f(x)) \tag{1}$$

where $h(x)$ is the hash value of entity x and $h(f(x))$ is the hash value of fingerprint $f(x)$. The two locations $i_1$ and $i_2$ are candidate storage locations to increase the flexibility of the lookup and insertion operations.

---

**Algorithm 2:** Entity Insertion Algorithm

---

$f(x)$ = fingerprint(x);
**if** *either bucket[$i_1$] or bucket[$i_2$] has an empty entry* **then**
    add $f(x)$ and the initialized temperature t to that bucket;
    add the addresses of the entity as block linked list to that bucket;
    **return** *True*
**end**
randomly select $i_1$ or $i_2$ as i;
**for** $k = 0; k < MaxNumKicks; k + +$ **do**
    select an entry m from bucket[i] randomly;
    interchange $f(x)$ and the fingerprint stored in entry m;
    i = i $\oplus$ hash($f(x)$);
    **if** *bucket[i] has an empty entry* **then**
        add $f(x)$ and the initialized temperature t to bucket[i];
        add the addresses of the entity as block linked list to bucket[i];
        **return** *True*
    **end**
**end**
**return** *False*

---

Then we check if there are empty slots at locations $i_1$ and $i_2$. If an empty slot exists in either position, the information of entity x including its fingerprint $f(x)$, its temperature t and its addresses are inserted into that empty slot and the insertion process is completed. If both locations are occupied, the Cuckoo Filter activates the eviction mechanism and tries to free up space for inserting a new entity by evicting the existing entity. In practice, the Cuckoo Filter sets the maximum number of evictions to avoid infinite recursive eviction operations; after reaching the maximum number of evictions, it means that the insertion operation fails.

### B. ENTITY EVICTION STRATEGY

In the eviction mechanism, the Cuckoo Filter randomly selects a location from $i_1$ or $i_2$ to evict the fingerprints therein and then calculate the alternative position j of the fingerprint. Next, it inserts the fingerprint into the empty slot of j. If it is already occupied, the eviction operation is repeated until an empty location is found or the maximum number of iterations is reached.

---

**Algorithm 3:** Entity Eviction Algorithm

---

$f(x)$ =fingerprint(x);
$i_1$ = hash(x);
$i_2 = i_1 \oplus$ hash($f(x)$);
**if** *bucket[$i_1$] or bucket[$i_2$] has $f(x)$* **then**
    remove $f(x)$ together with its information from this bucket;
    **return** *True*
**end**
**return** *False*

---

Through the eviction mechanism, Cuckoo Filter is able to continue inserting new entities under high load conditions, avoiding the high misclassification rate caused by direct rejection of insertion.

## C. 1-HOP + EASY QUESTION

- **Question:** What causes Horner's syndrome?
- **Key Answer:** Paralysis of ocular sympathetic nerves.
- **Rare Entity:** ocular_sympathetic_nerves.

**Text-Based RAG**

| Retrieval Time | Response Time | Time Ratio | Accuracy |
|---|---|---|---|
| - | 25.87s | - | 56% |

- **Question:** What causes Horner's syndrome?
- **Answer:** Horner's syndrome is caused by a disruption of the sympathetic nerve pathway that supplies the eye and face. This disruption can occur at any point along the three-neuron sympathetic chain, which extends from the hypothalamus to the eye...

**ANN Graph-RAG**

| Retrieval Time | Response Time | Time Ratio | Accuracy |
|---|---|---|---|
| 7.53s | 24.30s | 30.99% | 60% |

- **Question:** What causes Horner's syndrome?
- **Answer:** Symptoms of Horner's syndrome include ptosis (drooping eyelid), miosis (constricted pupil), anhidrosis (lack of sweating), and enophthalmos (sunken eye appearance). Identifying the underlying cause is crucial, as some causes, such as carotid artery dissection, require urgent medical attention...

**Cuckoo Filter Tree-RAG**

| Retrieval Time | Response Time | Time Ratio | Accuracy |
|---|---|---|---|
| 4.72s | 21.54s | 21.92% | 66% |

- **Relation:** Horner - ocular sympathetic nerves
- **Question:** What causes Horner's syndrome?
- **Answer:** Horner's syndrome is caused by paralysis of the ocular sympathetic nerves. Because Horner's syndrome can be a sign of serious conditions like carotid artery dissection or lung tumors, it often requires urgent medical evaluation...

## D. 2-Hop + Easy Question

- **Question:** What is the most essential component for culturing Mycoplasma?
- **Key Answer:** Cholesterol supplements.
- **Rare Entity:** cholesterol_supplements.

**Text-Based RAG**

| Retrieval Time | Response Time | Time Ratio | Accuracy |
|---|---|---|---|
| - | 25.12s | - | 45% |

- **Question:** What is the most essential component for culturing Mycoplasma?
- **Answer:** The most essential component for culturing Mycoplasma is a specialized growth medium containing sterols. Since Mycoplasma lack a cell wall, they require sterols to maintain membrane integrity...

**ANN Graph-RAG**

| Retrieval Time | Response Time | Time Ratio | Accuracy |
|---|---|---|---|
| 7.98s | 23.46s | 34.01% | 61% |

- **Question:** What is the most essential component for culturing Mycoplasma?
- **Answer:** The most essential component for culturing Mycoplasma is a rich growth medium supplemented with sterols. Since Mycoplasma species lack a cell wall and cannot synthesize sterols on their own, they require cholesterol for membrane stability...

**Cuckoo Filter Tree-RAG**

| Retrieval Time | Response Time | Time Ratio | Accuracy |
|---|---|---|---|
| 3.86s | 19.62s | 19.67% | 67% |

- **Relation:** Mycoplasma - cholesterol*, cholesterol* - horse serum, cholesterol* - cholesterol supplements
- **Question:** What is the most essential component for culturing Mycoplasma?
- **Answer:** The most essential component for culturing Mycoplasma is cholesterol. Mycoplasma are prokaryotic organisms that lack a cell wall, and their cell membrane requires cholesterol for stability and function. Therefore, the culture medium must include cholesterol, such as horse serum or cholesterol supplements, to support their growth. Additionally...

## E. Multi-Hop + Hard Question

When the problem is complex and the required entity relationships are precise, Tree-RAG shows an obvious advantage over the other two methods.

- **Question:** Here are five types of hydrogen carriers involved in electron transfer, which proceed in a specific order. Can you describe the detailed process?
- **Answer:** flavoprotein, coenzyme I, iron-sulfur protein, ubiquinone, cytochrome.
- **Rare Entity:** flavoprotein, coenzyme I, iron-sulfur_protein, ubiquinone, cytochrome.

**Text-Based RAG**

| Retrieval Time | Response Time | Time Ratio | Accuracy |
|---|---|---|---|
| - | 38.62s | - | 40% |

- **Question:** There are five types of hydrogen carriers involved in electron transfer, which proceed in a specific order. Can you describe the detailed process?
- **Answer:** NADH donates electrons to Complex I, FADH$_2$ donates to Complex II. Electrons transfer to ubiquinone (CoQ) converted to ubiquinol (QH$_2$). Ubiquinol transfers electrons to cytochrome b-c$_1$ complex (Complex III). Electrons pass through cytochrome c and reach Complex IV...

**ANN Graph-RAG**

| Retrieval Time | Response Time | Time Ratio | Accuracy |
|---|---|---|---|
| 9.02s | 30.15s | 29.91% | 49% |

- **Question:** There are five types of hydrogen carriers involved in electron transfer, which proceed in a specific order. Can you describe the detailed process?
- **Answer:** NADH donates electrons to Complex I (via FMN) or FADH$_2$ donates to Complex II. Fe-S clusters pass electrons one at a time to downstream carriers. Ubiquinone (CoQ) accepts electrons and protons, becoming ubiquinol (QH$_2$), which shuttles electrons to Complex III. Cytochromes in Complex III pass electrons to cytochrome c, which carries them to Complex IV. Complex IV (cytochrome c oxidase) transfers electrons to O$_2$, reducing it to H$_2$O...

**Cuckoo Filter Tree-RAG**

| Retrieval Time | Response Time | Time Ratio | Accuracy |
|---|---|---|---|
| 4.75s | 27.02s | 17.58% | 69% |

- **Relation:** electron transfer - hydrogen carriers, hydrogen carriers - process*, process* - 1.flavoprotein, process* - 2.coenzyme I, process* - 3.iron-sulfur protein, process* - 4.ubiquinone, process* - 5.cytochrome
- **Question:** There are five types of hydrogen carriers involved in electron transfer, which proceed in a specific order. Can you describe the detailed process?
- **Answer:** The process can be summarized as follows: flavoprotein → coenzyme I → iron-sulfur protein → ubiquinone → cytochrome. Flavoproteins, such as NADH dehydrogenase (Complex I) and succinate dehydrogenase (Complex II), contain flavin mononucleotide (FMN) or flavin adenine dinucleotide (FAD) as prosthetic groups. These enzymes play a crucial role in electron transfer from NADH or succinate to downstream components of the ETC...

## F. OFFLINE CONSTRUCTION COST

The complexity of Entity Extraction is linear $O(N)$ with respect to the number of documents in the corpus, as each document is processed independently by the LLM (or smaller IE models). While extraction is the most expensive step, it is a one-time offline cost. The Tree Construction and Cuckoo Filter insertion are negligible ($O(M)$ where $M$ is the number of entities) compared to the extraction.

## G. THE USE OF LLMS

We use llm to check and correct grammar and spelling mistakes. In addition, we also use llm to polish the sentences in our paper to make them more fluent.

