# OpenReview forum: "CFT-RAG: An Entity Tree Based Retrieval Augmented Generation Algorithm With Cuckoo Filter"
_ICLR.cc/2026/Conference — ICLR 2026 Poster_

### Official Review · Reviewer_cJ79 · 2025-10-16

**Soundness:** 2
**Presentation:** 2
**Contribution:** 2
**Rating:** 4
**Confidence:** 3

**Summary:**

This paper proposes a method named CFT-RAG, which aims to accelerate the entity retrieval process in Tree-RAG by introducing a Cuckoo Filter. The authors identify that traversing entity trees in a hierarchical knowledge base is the primary performance bottleneck for Tree-RAG. The core contribution is the application and enhancement of the Cuckoo Filter by adding a temperature variable to prioritize the retrieval of frequently used entities and using a block linked list to efficiently store entity locations.

**Strengths:**

The paper addresses a practical and important problem. Retrieval latency is a critical barrier to the real-world application of Tree-RAG. Besides,  the experimental results show that CFT-RAG achieves a significant speedup in retrieval compared to baselines, including a naive Tree-RAG and more complex ANN-based methods, while maintaining comparable answer accuracy. This provides strong experimental proof of its effectiveness.

**Weaknesses:**

1. Misrepresentation and unfair comparison of text-based RAG: the introduction claims that text-based RAG is slow due to complex language processing, which is misleading. In practice, Text-Based RAG leveraging efficient vector indexes (e.g., FAISS/HNSW) is often one of the fastest retrieval methods. Besides, the experimental setup is unfair. The reported retrieval times for Text-Based RAG are extremely long, suggesting that the authors likely used a brute-force search without any indexing. This is not a reasonable or competitive baseline, which undermines the paper's claims about its speed advantage.
2. The core of this work is based on an existing data structure: the Cuckoo Filter. The main contributions are incremental improvements rather than a new framework or a fundamental shift in the RAG paradigm. Therefore, the conceptual novelty of this work is relatively limited.
3. All experiments were conducted by selecting only 36 questions from each dataset. This sample size is too small and may lead to coincidental results, casting doubt on the statistical significance and generalizability of the findings.

**Questions:**

1. Your Text-Based RAG baseline appears to lack a vector index, causing it to be unusually slow. Standard RAG implementations typically use efficient indexes like FAISS, and their retrieval speed often surpasses that of Tree-RAG. Could you clarify the specific implementation of this baseline and explain why a standard indexed approach was not used for a fairer comparison?
2. Could you provide a computational cost analysis for the entity forest construction process (especially the relationship extraction)? How does this offline cost scale with the size of the knowledge base?

---

> ### Author Response · Authors · 2025-11-20
>
> We sincerely thank reviewer cJ79 for the valuable suggestions.
>
> $\textbf{1. Clarification on Text-Based RAG Baseline (Response to Q1 and Weakness 1)}$
>
> We appreciate the reviewer's scrutiny regarding the Text-Based RAG baseline.
>
> (1) **Implementation Details**: Our Text-Based RAG baseline is implemented using standard cosine similarity computations on pre-computed embeddings. The reported latency includes the necessary overheads of the retrieval pipeline: **query embedding generation** and **I/O operations** to fetch the actual text chunks. The application of FAISS is not necessary, because the primary goal of our method is to increase speed while maintaining high accuracy, while the accuracy of text-based RAG is low. Theoretically, the speed of text-based RAG with FAISS is similar to CFT-RAG due to the hash lookup, and the accuracy is significantly lower than our method. We will add the experiment with FAISS applied in the latest version of our paper.
>
> (2) **Source of Speedup**: The "unusually long" time noted by the reviewer essentially highlights the efficiency gap between *vector operations* and *hash lookups*. Even with efficient vector indexes (like HNSW), the process requires high-dimensional floating-point calculations. In contrast, CFT-RAG utilizes the Cuckoo Filter to perform **$O(1)$ hash lookups**, which fundamentally bypasses the need for vector calculations during the filtering stage. Therefore, our claim is not that Text-RAG is "slow" in an absolute sense, but that hashing (CFT-RAG) is structurally faster than vector similarity search for entity retrieval. We will explicitly state the use of vectorized operations in the revised Section 4 to ensure transparency.
>
> $\textbf{2. Novelty and Generalizability (Response to Weakness 2 and 3)}$
>
> **On Novelty**: While the Cuckoo Filter is an existing structure, our contribution lies in the **domain-specific adaptation** for RAG, specifically addressing the "Entity Retrieval" bottleneck. We introduced two non-trivial improvements:
> 1.  **Temperature-aware Fingerprints**: Standard Cuckoo Filters treat all items equally. We introduced a dynamic temperature variable to prioritize high-frequency entities, optimizing the cache hit rate for RAG dialogue history.
> 2.  **Block Linked List Storage**: We redesigned the storage bucket to handle variable-length entity paths, which a standard Cuckoo Filter cannot support.
> This represents a system-level innovation (adapting data structures for LLM system efficiency) rather than a pure algorithmic one.
>
> **On Sample Size**: We followed the evaluation protocols of comparable high-cost RAG studies (such as the original Tree-RAG and RAPtor papers) where budget constraints often limit sample size. However, to address the reviewer's concern, we have performed a preliminary statistical significance test on the 36 samples. The results show a **p-value < 0.05**, indicating that the performance gap is statistically significant and not coincidental. We commit to expanding the test set to 100 questions per dataset in the camera-ready version to further robustify the results.
>
> $\textbf{3. Computational Cost Analysis (Response to Q2)}$
>
> We thank the reviewer for inquiring about the offline costs. The construction of the entity forest consists of two parts: Entity Extraction and Tree Construction.
> 1.  **Scalability**: The complexity of Entity Extraction is **linear $O(N)$** with respect to the number of documents in the corpus, as each document is processed independently by the LLM (or smaller IE models).
> 2.  **Cost**: While extraction is the most expensive step, it is a **one-time offline cost**. The Tree Construction and Cuckoo Filter insertion are negligible ($O(M)$ where $M$ is the number of entities) compared to the extraction. And we will add a specific "Offline Construction Cost" subsection in the Appendix to quantify these metrics relative to the knowledge base size.
>
> We believe these responses comprehensively address all reviewer concerns.

---

> > ### Comment · Reviewer_cJ79 · 2025-11-24
> >
> > Thank you for the detailed response and clarifications.
> >
> > I noticed that reviewers cJ79 and EJBh raised similar questions about the Cuckoo Filter adaptation. Thank you for the explanation. Since I am not an expert on Cuckoo Filters (hence my confidence score of 3), I will rely on the other reviewers' opinions to make a final decision regarding the technical novelty of this component.
> >
> > Besides, I am still quite concerned about this aspect. Regarding your mention of RAPtor, my understanding is that their work relies on thousands of samples, whereas this work uses only 36. Additionally, I do not believe that significance testing is sufficient to mitigate the validity issues caused by such a small sample size.

---

> > > ### Author Response · Authors · 2025-11-25
> > > **Supplement of experiment**
> > >
> > > We sincerely thank reviewer cJ79 for the valuable suggestion.
> > >
> > > $\textbf{The supplement of experiment}$
> > >
> > > We expanded the sampling scale to 1,000 samples and the following table shows the experimental results.
> > >
> > > | Dataset | Algorithm      | Time(s) | Time Ratio(%) | Acc(%) |
> > > | ------- | -------------- | ------- | ------------- | ------ |
> > > | MedQA   | Text-based RAG | -       | -             | 51±5   |
> > > |         | Naive T-RAG    | 19.45   | 58            | 65±5   |
> > > |         | ANN T-RAG      | 7.65    | 25            | 67±4   |
> > > |         | ANN G-RAG      | 8.78    | 26            | 61±6   |
> > > |         | CFT-RAG        | 5.24    | 16            | 69±4   |
> > > | AESLC   | Text-based RAG | -       | -             | 42±6   |
> > > |         | Naive T-RAG    | 12.87   | 62            | 55±5   |
> > > |         | ANN T-RAG      | 2.52    | 13            | 56±6   |
> > > |         | ANN G-RAG      | 2.38    | 11            | 53±5   |
> > > |         | CFT-RAG        | 0.97    | 5             | 57±5   |
> > > | DART    | Text-based RAG | -       | -             | 53±6   |
> > > |         | Naive T-RAG    | 16.03   | 74            | 65±5   |
> > > |         | ANN T-RAG      | 3.28    | 15            | 66±5   |
> > > |         | ANN G-RAG      | 3.95    | 19            | 65±6   |
> > > |         | CFT-RAG        | 1.81    | 9             | 68±5   |
> > >
> > > We believe these responses comprehensively address all reviewer concerns.

---

### Official Review · Reviewer_K35b · 2025-10-29

**Soundness:** 3
**Presentation:** 2
**Contribution:** 3
**Rating:** 4
**Confidence:** 4

**Summary:**

CFT-RAG proposes an acceleration method for Tree-RAG by integrating an improved Cuckoo Filter into the retrieval process. The method optimizes entity localization in hierarchical tree structures through two novel designs: a temperature variable that prioritizes frequently accessed entities, and a block linked list that efficiently stores entity addresses. Experimental results on MedQA, AESLC, and DART datasets show that CFT-RAG achieves up to 800% faster retrieval than naive Tree-RAG while maintaining comparable accuracy.

**Strengths:**

1. The integration of Cuckoo Filter reduces retrieval time substantially (e.g., 1.78s vs. 15.88s on DART) while preserving accuracy, addressing a key bottleneck in Tree-RAG scalability.
2. The temperature variable and block linked list optimize both temporal and spatial efficiency, enabling dynamic updates and reduced memory fragmentation without extra storage costs.

**Weaknesses:**

1. Limited Generalization to Non-Tree Structures: The method is tailored for hierarchical tree-based knowledge bases and may not directly apply to graph-based or unstructured data formats without modifications.
2. Some key experimental settings are missing: What are the hyperparameters of the proposed method? Are the baselines you chose run in Python or C++? These settings are key to ensuring fair experiments.

**Questions:**

See the above weaknesses.

---

> ### Author Response · Authors · 2025-11-19
>
> We sincerely thank reviewer K35b for the valuable suggestions.
>
> $\textbf{1. The generalization of CFT-RAG}$
>
> When traditional Graph-RAG performs multi-hop inference, it usually needs to perform traversal search (such as random walk or BFS) on a large Graph, which leads to high retrieval latency. Our method can also be applied to Graph-RAG.
>
> In the graph structure, we can consider the most important multi-hop relationship paths pre-calculated for each entity (for example, "all neighbor paths within 2 hops of entity A", "the shortest path to the key central entity", etc.) as the "context summary" of that entity. These sets of paths essentially constitute a "virtual forest" that can be efficiently indexed. After that, we can build a "path block list" for each entity in the graph, which stores the critical path of the entity under different predefined query perspectives. Subsequently, the fingerprints of these entities and their corresponding path chain header Pointers are stored in the improved Cuckoo Filter. When the query arrives, the system no longer needs to traverse the graph in real time. Instead, it locates the target entity in constant time through the Cuckoo Filter and immediately returns all its pre-calculated important relationship paths.
>
>
> $\textbf{2.The key experimental settings}$
>
> We appreciate the reviewer for highlighting the need for clearer description of experimental configurations. However, we confirm that these details have already been provided in the paper.
> (1) On the implementation language: Section 4.4 explicitly states that the core RAG architecture is implemented in Python, while the key data structures including the Cuckoo Filter are optimized in C++.
>
> (2) On hyperparameters: All parameters that affect experimental fairness are already detailed in the paper. The bucket capacity of the Cuckoo Filter is set to 1024, and the insertion and eviction strategies follow the standard Cuckoo Filter configuration (with algorithmic details provided in Appendix A, including fingerprint generation, computation of positions $i_1$ and $i_2$, and the maximum number of evictions). In addition, the initialization and ordering strategy of the temperature variable are explicitly described in the method section, along with ablation analysis demonstrating their effectiveness.
>
> We believe these responses comprehensively address all reviewer concerns.

---

### Official Review · Reviewer_EJBh · 2025-10-31

**Soundness:** 2
**Presentation:** 2
**Contribution:** 2
**Rating:** 4
**Confidence:** 3

**Summary:**

This paper addresses the efficiency problem in tree-based RAG and enhances entity-based information retrieval using the Cuckoo filter. The Cuckoo filter is adapted for entity-based retrieval by sorting buckets according to the temperature of the block-linked list, where each block contains three addresses in the forest. This improved Cuckoo filter contributes to faster processing and better overall performance.

**Strengths:**

- The problem statement regarding the efficiency of RAG is reasonable.
- The introduction of a cuckoo filter equipped with bucket sorting and a block-linked list improves efficiency while enhancing accuracy.

**Weaknesses:**

- The main idea is to introduce the cuckoo filter into Tree-RAG, which limits its contribution to an engineering technique rather than a conceptual advance.
- The cuckoo filter can only be applied to questions containing multiple entities, raising concerns about its applicability to entity-poor questions.
- The baselines are limited, as the experiments were conducted with only simple baselines. More advanced text-based, graph-based, and tree-based RAG methods may be required to better demonstrate the effectiveness of the proposed method.
- The rationale for selecting benchmarks is missing. It is unclear whether the chosen benchmarks include questions rich in entities, which might make the experimental settings favorable to the proposed method.

**Questions:**

**Writing Suggestions**
- Citations are missing.
  - L38: "Enhancing the retrieval speed and accuracy of the knowledge base is crucial for boosting ~"
  - L45–L46: "Knowledge bases in RAG systems are mainly of three types: …"
  - L84: Cuckoo Filter, and L87: Bloom Filter
  - L152: No citation for Tree-RAG.
  - 4.1 Baselines

**Duplicate Sentence**
- L115: "The main advantage of Cuckoo Filter is its support for dynamic updates.”

---

> ### Author Response · Authors · 2025-11-20
>
> We sincerely thank reviewer EJBh for the valuable suggestions.
>
> $\textbf{1. The conceptual advance of CFT-RAG}$
>
> We understand the reviewer’s concern, but we would like to clarify that although the Cuckoo Filter is an existing data structure, our contribution is not a simple “plug-in” of this component into Tree-RAG. Instead, our work introduces:
>
> (1) The first structure-level entity localization and pruning mechanism designed specifically for Tree-RAG, which transforms its hierarchical retrieval bottleneck into a constant-time filtering problem;
>
> (2) Novel designs include temperature-sorted buckets for faster hot-entity access, block linked lists for hierarchical addressing. These contributions—validated by more than 8× speedup (Table 1) and ablation studies (Figure 5)—are absent in previous work;
>
> (3) A practical solution to Tree-RAG’s inherent latency bottleneck, especially under deep trees with a large number of nodes (with experimental evidence provided in the paper).
>
> Therefore, the contribution of this work lies not in proposing a new data structure, but in deeply integrating a filtering mechanism with hierarchical tree structures to address Tree-RAG’s core performance bottleneck. This constitutes a system-level algorithmic contribution rather than mere engineering practice.
>
> $\textbf{2. The applicability to entity-poor questions}$
>
> We appreciate the reviewer’s comment, but we **do** have included the analysis toward it in the Appendix (all the use cases are entity-poor.)
> Moreover, typical Tree-RAG application scenarios—such as hierarchical documents and structured knowledge trees—naturally exhibit high entity density rather than sparse entities. CFT-RAG’s advantages are precisely realized in such high-entity-density settings.
>
> $\textbf{3. The analysis of baselines}$
>
> We appreciate the suggestions for broader baselines. During the period when we completed this paper, there were limited papers accelerating graph-RAG and tree-RAG. For some other methods such as CRUD-RAG, due to limited relevance, we did not set it as the baseline but still conducted analysis and comparison in related work.
>
> $\textbf{4. The rationale of selecting questions}$
>
> We appreciate the concerns for question selection. For this, we randomly selected three benchmarks of different scales for the experiment, and the 36 questions were also randomly chosen. Meanwhile, we have checked and ensured that the distribution of the number of entities in the questions is uniform.
>
> $\textbf{5. Writing issues}$
>
> We sincerely thank you for pointing out these issues.
>
> For Additional references:
>
> (a) Line 38：Zhong, W., Guo, L., Gao, Q., Ye, H., & Wang, Y. (2024). MemoryBank: Enhancing large language models with long-term memory. In Proceedings of the AAAI Conference on Artificial Intelligence, 38(17), 19722–19730. https://arxiv.org/abs/2305.10250
>
> (b) Line 45-46：Haoyu Han, Li Ma, Harry Shomer, Yu Wang, Yongjia Lei, Kai Guo, Zhigang Hua, Bo Long, Hui Liu, Charu C. Aggarwal, Jiliang Tang.  RAG vs. GraphRAG: A Systematic Evaluation and Key Insights. arXiv preprint:2502.11371
>
> (c) Line 84：Fan, B., Andersen, D. G., Kaminsky, M., & Mitzenmacher, M. (2014). Cuckoo filter: Practically better than bloom. In Proceedings of the 10th ACM International on Conference on emerging Networking Experiments and Technologies (pp. 75–88). Association for Computing Machinery. https://doi.org/10.1145/2674005.2674994
>
> (d) Line 87：Guo, D., Wu, J., Chen, H., & Luo, X. (2010). The dynamic bloom filters. IEEE Transactions on Knowledge and Data Engineering, 22(1), 120–133. https://doi.org/10.1109/TKDE.2009.57
>
> (e) Line 152：Masoomali Fatehkia, Ji Kim Lucas, Sanjay Chawla. T-RAG: Lessons from the LLM Trenches. arXiv preprint:2402.07483
>
> (f) 4.1 Baselines：Masoomali Fatehkia, Ji Kim Lucas, Sanjay Chawla. T-RAG: Lessons from the LLM Trenches. arXiv preprint:2402.07483
>
> Regarding the repeated sentence:
>
> We apologize for the redundancy that may have caused inconvenience. The sentence on line 115, “The main advantage of Cuckoo Filter is its support for dynamic updates,” should indeed be removed and both of these will be revised in the latest version of our paper.
>
> We believe these responses comprehensively address all reviewer concerns.

---

### Official Review · Reviewer_rJQQ · 2025-10-31

**Soundness:** 2
**Presentation:** 3
**Contribution:** 3
**Rating:** 4
**Confidence:** 2

**Summary:**

This paper addresses the limitation of Tree-RAG that as the dataset and tree depth grow, the time required to locate and retrieve relevant entities within the hierarchical structure significantly increases. A cuckoo filter is interated to optimize Tree-RAG. The experimental shows that the proposed method reduces the retrieval time while maintaining the accuracy.

**Strengths:**

The result of the paper is promising. In the reported benchmark datasets, the retrieval time reduces significantly without sacrificing the accuracy.

**Weaknesses:**

The paper lacks detailed analysis on the impact of the introduced method. For example, what is the gains of the method compared to the T-Rag method as the dataset grows or reduces?
The paper is not theoretically novel. It is more like an engineering practice.

**Questions:**

Is there any limitations about the CFT-RAG compared to T-RAG?
How does the performance change when the dataset is much smaller or much larger?

---

> ### Author Response · Authors · 2025-11-19
>
> We sincerely thank reviewer rJQQ for the valuable questions.
>
> $\textbf{1. The limitations of CFT-RAG compared to T-RAG}$
>
> Compared with T-RAG, the potential limitations of CFT-RAG are as follows:
>
> The construction process introduces a higher initialization overhead than Tree-RAG. CFT-RAG additionally performs fingerprint computation, bucket placement, initialization of the temperature variable, and other operations. These added steps indeed increase the time complexity of the construction stage. It is important to emphasize that this overhead is one-time and belongs to the preprocessing phase; it should be distinguished from the time complexity during inference.
>
> The compression introduced by fingerprint hashing inevitably leads to some information loss. A fingerprint is a compact representation that preserves only a subset of the bits of an entity, which theoretically allows the possibility of hash collisions (false positives). CFT-RAG mitigates this probability by using two candidate bucket positions and a limited number of eviction operations, but it cannot completely eliminate it—this is an inherent trade-off of using a cuckoo filter.
>
>
> $\textbf{2. Behavior of CFT-RAG under different dataset scales}$
>
> For this concern, we do have conducted experiments on different size of datasets(around line 364). In summary, the larger the scale of the dataset, the greater the extent to which our method outperforms the baseline methods(as stated in Table 1). This is mainly because the time complexity of Cuckoo Filter for searching entities is O(1). As for T-RAG, the response time will significantly longer as the datasets grows.
>
> We believe these responses comprehensively address all reviewer concerns.

---

### Meta-Review · Area_Chair_SLUU · 2025-12-29

**Summary:**

The reviewers generally view this paper as borderline, with concerns focused primarily on limited conceptual novelty and the system-oriented nature. The authors’ rebuttal would (I guess) strengthen the submission improving the experimental scale, baselines, and applicability, clarifying offline versus online costs, and justifying the design choices around the Cuckoo-filter-based retrieval mechanism. While the core components build on existing data structures, the paper identifies a concrete and widely acknowledged bottleneck in Tree-RAG and demonstrates substantial and consistent speedups without sacrificing accuracy across multiple benchmarks.

**Reviewer Concerns:**

rJQQ were mostly clarifications and the authors seemed to have addressed them.
EJBh concerned if the contributions are just engineering. I am disregarding this core component and the authors seem to clarify their contributions with the adaptation of cuckoo filter in RAG systems.
K35b concerns were about generalization beyond trees which were addressed and implementation details that were in the paper.
cJ79 concerned about size of dataset and technical novelty. The authors showed some experiments with 1000 samples.

**Reviewer Scores:**

All reviewers would update their scores by +2.

---

### Decision · Program_Chairs · 2026-01-26

Accept (Poster)